# Effects of Orally Delivered Double-Stranded RNA of *Trehalose-6-Phosphate Synthase* on the Population of *Frankliniella occidentalis*

**DOI:** 10.3390/insects16060614

**Published:** 2025-06-10

**Authors:** Tao Lin, Xiaoyu Chen, Ying Chen, Ting Chen, Xueyi Liang, Hui Wei, Guang Yang

**Affiliations:** 1State Key Laboratory of Agricultural and Forestry Biosecurity, Institute of Applied Ecology, Fujian Agriculture and Forestry University, Fuzhou 350002, China; maludongzuo@163.com (T.L.); 18305988226@163.com (X.C.); cyay132114@163.com (Y.C.); xueyiliang2022@163.com (X.L.); 2Fujian Engineering Research Center for Green Pest Management, Fujian Key Laboratory for Monitoring and Integrated Management of Crop Pests, Institute of Plant Protection, Fujian Academy of Agricultural Sciences, Fuzhou 350013, China; ct318815@163.com; 3International Joint Research Laboratory of Ecological Pest Control, The Innovation and Talent Recruitment Base of Ecological Control of Subtropical Crop Pests, International Cooperation Base for Science and Technology on Ecological Control of Crop Pests in Fujian and Taiwan, Ministerial and Provincial Joint Innovation Centre for Safety Production of Cross-Strait Crops, Fujian Agriculture and Forestry University, Fuzhou 350002, China; 4Key Laboratory of Integrated Pest Management for Fujian-Taiwan Crops, Ministry of Agriculture and Rural Affairs, Fuzhou 350002, China

**Keywords:** feeding, trehalose-6-phosphate synthase, life table, population suppression, bacterially expressed dsRNA, western flower thrip

## Abstract

Western flower thrips (WFTs, *Frankliniella occidentalis*) are damaging invasive pests in agriculture. Traditional chemical controls have led to resistance and environmental concerns, driving the search for eco-friendly alternatives. RNA interference (RNAi) targeting the *trehalose-6-phosphate synthase* (*TPS*) gene shows potential for pest management. Feeding dsRNA targeting *FoTPS* significantly increased the mortality of WFT nymphs and adults and reduced the egg production of adults. Using bacteria to deliver dsFoTPS further extended the pre-reproductive period and decreased survival, fecundity, and population growth. This study demonstrates that RNAi targeting *FoTPS* can effectively suppress WFT populations, offering a sustainable pest control strategy.

## 1. Introduction

The western flower thrip (WFT), *Frankliniella occidentalis* (Pergande), is a pervasive agricultural and horticultural pest with global economic implications [1]. WFTs were first discovered in California in 1895 and had been confined to western North America until the 1960s, subsequently rapidly spread globally through the horticultural trade, and then gradually spread and became a major pest affecting many crops worldwide. WFTs are predominant in both field and greenhouse crops, and the transmission of viruses exacerbates the economic losses in addition to their direct damage from feeding and oviposition [2,3]. Currently, WFTs are distributed in at least 57 countries worldwide, posing a severe challenge to integrated pest management (IPM) strategies owing to their high adaptability [2]. Nevertheless, in the absence of sustainable management strategies, chemical pesticides remain the predominant method for WFT control [4]. The indiscriminate use of pesticides has led to the development of resistance in WFTs, while concurrently depleting their natural enemies and competitors, thereby amplifying control challenges [5]. It is imperative to explore innovative pest management measures to decrease or replace the use of chemical pesticides.

RNA interference (RNAi) is a promising alternative to chemical pesticides [6]. It is typically employed to reduce or silence the expression of target genes, which can impede the development of insect pests and potentially lead to their mortality, holding substantial potential for the development of novel pest management strategies [7]. Recently, the advent of RNAi-based plant protection products has further stimulated the rapid development of this new alternative measure [8,9]. However, variations in insect morphology, survival, development, reproduction, rhythmic behavior, sex determination, predation patterns, digestion, metabolism, and stress tolerance present significant challenges in identifying efficient target genes for RNAi-based pest control strategies [10,11]. Given that survival, development, and reproduction are critical for pest population proliferation, identifying key genes regulating these processes and optimizing RNAi efficiency are essential for effective pest control [7,12]. For instance, the microinjection of dsRNA targeting ATP synthase subunit B (V-ATPase-B) in female WFTs significantly impacts their survival and reproduction [13]. Similarly, the oral delivery of dsRNA via a leaf feeding system has enabled the identification of several lethal target genes in WFTs [14], while silencing the superoxide dismutase (SOD) gene through RNAi markedly reduces pupal emergence rates and adult fertility [15]. Despite these successes, the efficiency of naked dsRNA delivery remains limited [16]. Alternative strategies, such as transgenic plant- [17], symbiont- [18], and plastid-mediated [6] RNAi, have been investigated for the management and control of WFTs [13,19]. Notably, engineered microbial systems can produce large quantities of dsRNA at a low cost, enabling spray-induced gene silencing (SIGS) for crop application [20], thereby offering a promising approach for a scalable and efficient RNAi-based management strategy for WFT control.

Trehalose, often referred to as the “blood sugar” of insects [21], is pivotal in all developmental stages, serving as an energy reserve and cryoprotectant under stressful environments [22]. Trehalose is a disaccharide consisting of two glucose molecules linked by a glycosidic linkage [23]; it is predominantly synthesized in the fat body of insects and transported into the hemolymph and other tissues via the trehalose transporter [24]. It is ultimately cleaved into two glucose moieties by the trehalase enzyme (TRE). Trehalose is primarily biosynthesized in insects via the trehalose-6-phosphate synthase (TPS)/trehalose-6-phosphate phosphatase (TPP) pathway [25]. In this pathway, TPS transfers glucose from UDP-glucose to glucose-6-phosphate, yielding trehalose-6-phosphate and UDP [26,27]. As a key enzyme in the trehalose biosynthesis pathway in insects, TPS plays an integral role in various biological processes, including insect development, energy metabolism, metamorphosis, stress recovery, chitin synthesis, and flight [28]. Silencing the *TPS* gene significantly increases the rate of molting deformities in *Nilaparvata lugens* [22,29] and *Acyrthosiphon pisum* [30]; significantly decreases the survival rate of *Lissorhoptrus oryzophilus* [31], *Ostrinia furnacalis* [27], and *Heortia vitessoides* [32]; and inhibits larval development in *Diaphorina citri* [33,34]. However, the silencing effects of *TPS* have not yet been reported in WFTs.

In this study, we investigated the mortality and egg production of WFTs fed with in vitro synthetic dsFoTPS. Furthermore, through life-table experiments, we elucidated the suppression of the WFT population via the oral delivery of bacterially expressed dsFoTPS, thereby assessing the potential of RNAi-based strategies for WFT management.

## 2. Materials and Methods

### 2.1. Insect Rearing

In accordance with the methodology outlined by Lin et al. [35], the feeding system for WFT adults used a 0.5 L PVC container. Briefly, this container featured a bottom aperture that was sealed with a stretched film (Pechiney Plastic Packaging, Chicago, IL, USA) and capped with a nylon net lid to facilitate aeration. The container was then inverted and allowed to float in a water-filled dish. Approximately 200 adult thrips were introduced into the container, and germinated broad bean seeds were provided as food, which were replaced every 2–3 d. The water was subsequently decanted, and the WFT eggs were harvested by concentrating them on filter paper, and the stretched film was replaced. After hatching, the nymphs were nurtured on germinated broad bean seeds, and specimens at different developmental stages were collected for experimental purposes. Adults and nymphal stages were cultivated under a photoperiod of 16 h/8 h light/dark, a temperature of 22 ± 1 °C, and a relative humidity (RH) of 75% ± 5%.

### 2.2. RNA Extraction and cDNA Synthesis

Total RNA was extracted from WFT developmental stages, including egg, 1st- and 2nd-instar nymph, prepupa, pupa, and adult, by using the TRNzol Universal RNA reagent (Tiangen, Beijing, China), following the manufacturer’s protocol. Subsequently, the first-strand cDNA was synthesized via reverse transcription by using the FastKing gDNA Dispelling RT Super Mix Kit (Tiangen, Beijing, China). The resulting cDNA samples were subsequently stored at −20 °C for future study.

### 2.3. Synthesis of dsRNA

A sequence analysis of the conserved domains of the *FoTPS* gene (XM_026416906.2) in the WFT was performed in NCBI (https://blast.ncbi.nlm.nih.gov/Blast.cgi, assessed on 17 October 2023). Details regarding the primers used in this study are provided in Appendix A. Primers with the T7 polymerase promoter sequence (TAATACGACTCACTATAGGG) added to the 5′ ends were designed by using the Primer3Plus (https://www.primer3plus.com/). These primers and the Phanta Max Super-Fidelity DNA Polymerase (Vazyme, Nanjing, China) were used to amplify gene fragments, and the dsRNA synthesis templates were prepared according to the following procedure: an initial denaturation step of 30 s at 95 °C; 35 cycles of denaturation at 95 °C for 15 s, annealing at 53 °C for 15 s, and extension at 72 °C for 30 s; and a final extension at 72 °C for 7 min. Amplified fragments were separated in agarose gel and purified by using the E.Z.N.A Gel Extraction Kit (Omega, Bio-tek, Narcross, GA, USA) following the manufacturer’s instructions. The dsRNAs were synthesized using the T7 RiboMAX™ Express RNAi System Kit (Promega, Madison, WI, USA), following the manufacturer’s protocol, and then the dsRNA was purified using an RNA extraction solution (phenol/chloroform/isoamyl alcohol = 25:24:1, *v*/*v*) and the ethanol precipitation method, followed by elution in diethyl pyrocarbonate (DEPC)-treated nuclease-free water. The size of the dsRNA products was determined using agarose gel electrophoresis and quantified using a spectrophotometer (Nano Drop One C, Thermo Fisher, Waltham, MA, USA) after 10 rounds of dilution. The sample was stored at −80 °C for subsequent study. The enhanced green fluorescent protein (EGFP) gene was used as a control.

### 2.4. Bacterially Expressed dsRNAs

Bacterial dsRNA was expressed using the L4440-HT115(DE3) system. To construct L4440-TPS, a PCR-amplified fragment was enzymatically digested with *Xba*I and *Xho*I and subsequently cloned into the *Xba*I–*Xho*I restriction site of the L4440 vector (Appendix A). The recombinant L4440 vectors were then transformed into *Escherichia coli* HT115 (DE3) competent cells, which were RNase III-deficient, and cultured in a Luria–Bertani (LB) medium supplemented with ampicillin at a final concentration of 50 μg·mL^−1^, with vigorous shaking at 200 rpm overnight at 37 °C. An inoculum (1 mL) of this bacterial culture was transferred to the LB medium (100 mL) and incubated for approximately 3 h until the optical density (OD) reached 0.4. Isopropyl β-D-1-thiogalactopyranoside (IPTG) was then added, and the culture was vigorously shaken for 4 h to induce the expression of dsRNA. Total RNA was extracted from the bacterial solution by using the Trizol method to verify dsRNA expression (Appendix A). The recombinant HT115 strain expressing dsEGFP was obtained from the repository of our research group.

### 2.5. Effect of In Vitro-Synthesized dsFoTPS on WFTs

Initial experiments were conducted to assess the efficacy of dsFoTPS on the 1st- and 2nd-instar nymphs and adults of WFTs under the above-mentioned environmental conditions. A reduced-scale PVC container (28.4 mL) was used as the experimental unit. Broad beans of 5 g were immersed in a solution of 500 ng/mL dsFoTPS and dsEGFP for 1 min, air-dried for 5 min, and then introduced into the PVC container containing 30 first- or second-instar nymphs (molting within 24 h) or female adults (20 d old) of WFTs. The top opening of the PVC container was sealed by using a cotton plug to ensure adequate ventilation. The device, with its parafilm-covered end, was inverted and floated on a water tray. The broad beans soaked in the dsRNA solution were replaced every 24 h. This treatment was repeated five times for each instar nymph. The individual mortality and egg production were recorded at 24, 48, and 72 h. Concurrently, the same experimental treatment was carried out simultaneously for sampling. Surviving individuals (at a minimum of 10 WFTs at each sampling time, repeated three times) were sampled, and samples were collected, flash-frozen in liquid nitrogen, and stored at −80 °C for subsequent qPCR validation.

### 2.6. Construction of Life Table of WFT Population

The experimental setup and environmental conditions were consistent with those described above. Ten newly laid eggs of the WFTs were then selected and introduced into the PVC container. On the third day post-deposition, a small broad bean was immersed in a bacterial solution that was induced to express dsRNA for 1 min and then transferred to the container to provide food for the hatchling nymphs. Broad beans soaked in bacterial solution expressing dsRNA, along with new parafilm, were replaced every 24 h. Each treatment was repeated four times, with daily monitoring of mortality, developmental stage, and fecundity within the experimental unit until the death of all individuals. The same experimental treatment was carried out simultaneously for sampling, and surviving individuals (at a minimum of 10 WFTs at each sampling time, repeated three times) were collected on days 7, 16, and 28, flash-frozen in liquid nitrogen, and stored at −80 °C for subsequent qPCR validation.

### 2.7. Expression Level Analysis

Quantitative real-time PCR (qPCR) assays were conducted to quantify the expression level of *FoTPS* by utilizing the GoTaq^®^ qPCR MasterKit (Promega, Madiso, WI, USA) following the manufacturer’s guidelines on the Applied Quant Studio^TM^ 6 Pro Real-Time PCR system (Applied Biosystems, Thermo Fisher Scientific, Waltham, MA, USA). Each qPCR reaction was executed in a 20 µL volume comprising 7.2 µL of nuclease-free water, 10 µL Master Mix, 0.4 μL of each specific primer, and 2.0 µL of the cDNA template. The qPCR cycling parameters encompassed an initial step of denaturation at 95 °C for 2 min; 40 cycles of denaturation at 95 °C for 10 s and annealing for 30 s at 60 °C; and a melting step from 60 to 95 °C. The *β-actin* gene of WFT (XM_052268150.1) was used as the reference gene, and the relative quantification of *FoTPS* expression was determined using the 2^−ΔΔ*Ct*^ method.

### 2.8. Data Analysis

MEGA 6.06 software (https://www.megasoftware.net/history.php, assessed on 29 August 2020) was employed to analyze the gene families of reference species and their evolutionary relationships. The maximum likelihood method was utilized to construct the phylogenetic tree, with the branch confidence ensured through 1000 bootstrap replications. For the sequence alignment of target sequences, Clustal X (version 2.1) (http://www.clustal.org/clustal2/, assessed on 20 March 2022) was applied. Sequence similarity analysis was conducted between the WFT and its clade species using TBtools software, (TBtools-II v2.102), and the results were visualized via HeatMap. The physicochemical properties were obtained by using ExPASY (https://www.expasy.org/, assessed on 6 September 2024). Homology modeling was employed to predict the protein structure using the SWISS-MODEL (https://swissmodel.expasy.org, assessed on 20 July 2024). The three-dimensional structures of the proteins were visualized and compared using PyMol software (version 2.5) (https://www.pymol.org/, assessed on 22 July 2024). Additionally, the secondary structure of the proteins was predicted using ESPript (version 3.0) (https://espript.ibcp.fr/ESPript/cgi-bin/ESPript.cgi, assessed on 23 July 2024).

The expression levels of *FoTPS* across various developmental stages of WFTs were subjected to a one-way ANOVA (analysis of variance) with Tukey’s multiple-comparisons HSD test (*p* < 0.05). Mortality, the quantity of eggs, and *FoTPS* expression levels in the WFT after the ingestion of in vitro-synthesized dsRNAs were statistically evaluated by using Student’s *t*-test. The life-table data obtained from WFTs fed with bacteria expressing dsRNA were analyzed using the age-stage, two-sex life-table theory and the methodology detailed by Chi et al. [36]. The computational formulas for the diverse parameters are presented in Appendix A. These parameters were compared using a bootstrapping technique, which facilitated the estimation of means, variances, and standard errors through 100,000 bootstraps via the TWOSEX-MSChart program [37]. The TIMING-MSChart program [38] was used to predict the potential growth of WFT populations feeding on different bacteria expressing dsRNA. For comparative purposes, the population size and stage structure of WFTs were simulated by assuming unlimited growth from an initial population of 10 eggs for a 100-day period.

## 3. Results

### 3.1. Sequence Characteristics of FoTPS

The full-length cDNA sequence of *FoTPS* from WFTs consisted of nucleotides with 4072 bp, with an open reading frame (ORF) of 2478 bp encoding a protein with 825 AAs. The predicted molecular mass of the protein was 93.213 kDa, with a theoretical isoelectric point (pI) of 6.08. Phylogenetic analyses indicated that FoTPS was closer in affinity with TPSs of Hemiptera than with those of other taxonomic groups (Appendix A). A sequence similarity analysis demonstrated that *TPS* exhibited >80.00% identity with *FoTPS* in other reference insects (Appendix A). Multiple sequence alignment revealed that FoTPS had 484 highly conserved amino acids with AgTPS (*Aphidius gifuensis*), NlTPS (*N. lugens*), DmTPS (*Drosophila melanogaster*), MdTPS (*Musca domestica*), BmTPS (*Bactrocera minax*), and TdTPS (*Trichogramma dendrolimi*). The three-dimensional structures of FoTPS, AgTPS, NlTPS, and TdTPS were generated by molecular homology modeling, and their sequence consistencies were 100.00%, 78.72%, 99.50%, and 79.51%; in addition, the global model quality estimation (GMQE) values were 0.85, 0.87, 0.87, and 0.89, respectively (Appendix A). The MolProbity scores of these four proteins were 1.22, 1.27, 0.95, and 1.13, and Ramachandran plot values for favored regions were 93.44%, 95.13%, 95.90%, and 95.64%. The root-mean-square deviation (RMSD) values of FoTPS and AgTPS, NlTPS, and TdTPS were 1.457, 0.706, and 0.040, respectively (Appendix A). The analysis revealed that the FoTPS sequence exhibited high similarities with other insect TPSs, with conserved structural and physicochemical properties, indicating a similar function of FoTPS with other insect TPSs.

### 3.2. Expression Pattern of FoTPS Across Different Developmental Stages

A significant variation in the expression of *FoTPS* was observed across the developmental stages of WFTs (*F*_5,12_ = 1676.907, *p* < 0.001). The adult stage exhibited the highest expression level, followed by prepupa, egg, and second-instar larva stages in that order, and first-instar larva and pupa stages had the lowest expression levels (Figure 1).

### 3.3. Effects of Ingestion of Synthesized dsFoTPS on Mortality and Egg-Laying Behavior of WFTs

The first-instar nymphs of WFTs exhibited significant differences in mortality after ingesting in vitro-synthesized dsFoTPS compared with the individuals ingesting dsEGFP in 24, 48, and 72 h (Figure 2A). Similarly, for the second-instar nymphs, the mortality rates were significantly increased by dsFoTPS compared to dsEGFP in 24, 48, and 72 h. The ingestion of dsFoTPS led to 5.5- and 4.0-fold mortality rates in the second-instar nymphs compared with the ingestion of dsEGFP controls in 48 h and 72 h, respectively (Figure 2B). For the adults, no significant differences were observed in mortality in 24 h and 48 h, but the mortality rate was significantly increased by dsFoTPS compared with dsEGFP in 72 h (Figure 2C). Although no significant increase in fecundity was observed by the ingestion of dsFoTPS in WFT adults in 24 h and 48 h, there was a significant decrease in fecundity in 72 h (Figure 2D).

### 3.4. Effects of Ingestion of Synthesized dsFoTPS on Gene Expression in WFTs

The expression of *FoTPS* in WFTs was markedly down-regulated after the ingestion of dsFoTPS. In the first-instar larvae of WFTs, *FoTPS* expression was significantly down-regulated by 74.9%, 84.3%, and 80.1% at 24, 48, and 72 h, respectively (Figure 3). For the second-instar nymphs, *FoTPS* expression was significantly down-regulated by 75.2%, 90.5%, and 88.6% at 24, 48, and 72 h, respectively (Figure 3). In adult WFTs, *FoTPS* expression was significantly down-regulated by 91.8%, 87.9%, and 90.1% at 24, 48, and 72 h, respectively (Figure 3).

### 3.5. Effects of E. coli Expressing dsFoTPS on Life-Table Traits of WFT Population

The PCR identification of *E. coli* monoclonal colonies revealed that the positive clones were 856 bp (*TPS*) and 821 bp (*EGFP*). The dsRNA induced by IPTG in HT115 showed bright target bands at 223 bp and 255 bp, indicating the production of dsFoTPS and dsEGFP (Figure 4).

No significant variations were detected in the developmental duration of the eggs, nymphs, and pupae between WFTs fed with *E. coli* expressing dsFoTPS and the control group of WFTs fed with dsEGFP bacteria (Table 1). The developmental duration of preadults and the adult preoviposition period (APOP) of WFTs fed with *E. coli* expressing dsFoTPS was unchanged compared with that of the control group; however, the total preoviposition period (TPOP) was considerably increased (Table 1). The total longevity of female and male WFTs was considerably reduced when WFTs were fed with *E. coli* expressing dsFoTPS (*p* < 0.05), with only 50% of the cohort surviving to 31.9 d, which was significantly less than the 54.0 d observed in the control (*p* < 0.05). Despite no significant difference in the age at which WFTs fed with *E. coli* expressing dsFoTPS reached their reproductive peak compared to the control, both their fecundity and oviposition days were significantly shorter than the control (Table 1; Figure 5).

The superimposed age-stage-specific survival rate (*s_xj_*) clearly illustrates the variability in developmental rates among WFT individuals (Figure 5A,A’). The mortality rates of WFT nymphs feeding on bacterially expressed dsFoTPS significantly exceeded those of the control (*p* < 0.05) (Table 2; Figure 5B,B’). Although the age-stage-specific fecundity (*f_xj_*) of the WFTs gradually decreased with increasing age, three peaks were identified for the population feeding on bacterially expressed dsEGFP at 9.0 on day 21, 7.8 on day 41, and 7.4 on day 54, with age-stage-specific fecundity (*m_x_*) sustained in approximately three offspring from day 21 to day 49. However, the *m_x_* value of the WFT population fed with dsFoTPS only peaked (2.5) on day 22, with *m_x_* subsequently decreasing (Figure 5B,B’).

### 3.6. Effects of E. coli Expressing dsFoTPS on Population Parameters and Population Projection

The life expectancy (age 0) of WFTs fed with *E. coli*-expressed dsEGFP and dsFoTPS was 52.7 d and 29.9 d, respectively (Figure 5C,C’). The reproductive values (*v_xj_*) also corresponded to the finite rate of increase (*λ*) in the respective treatments (Table 1; Figure 5D,D’). Notably, the *v_xj_* value of both adult WFT treatments peaked on day 21 at 47.4 for dsEGFP and 26.0 for dsFoTPS. The intrinsic rate of increase (*r*), *λ*, net reproductive rate (*R*_0_), and mean generation time (*T*) of WFTs fed with bacterially expressed dsFoTPS were significantly reduced compared to those fed with dsEGFP (Table 1; Figure 5D,D’).

The simulations of population growth in 100 d indicated that the stage population sizes of WFTs fed with *E. coli* expressing dsFoTPS were 77,011 eggs, 62,086 nymphs, 10,786 pupae, 8726 females, and 7869 males. Those of dsEGFP were 2,928,148 eggs, 2,004,889 nymphs, 379,666 pupae, 217,553 females, and 178,170 males, which correspond to 38, 32, 35, 25, and 23 times for the dsFoTPS treatment. The total population size of WFTs fed with dsFoTPS bacteria (166,481) was 1/34 of that fed with dsEGFP bacteria (5,708,430) (Figure 5E,E’).

### 3.7. Effects of Ingestion of E. coli Expressing dsFoTPS on Gene Expression

The expression level of *FoTPS* in the WFTs was significantly down-regulated on day 7 (*p* < 0.001) and day 28 (*p* < 0.001) in the dsTPS treatments compared with that in the dsEGFP treatment, with gene expression being reduced by 2.4- and 4.7-fold, respectively. However, after 16 d of continuous exposure to dsFoTPS, no significant difference in expression level was observed when compared with the dsEGFP treatment (Figure 6), which may be due to the sampling period being during the pupal stage and no feeding behavior occurring.

## 4. Discussion

Trehalose synthesis does not occur in vertebrates, particularly mammals [39], rendering the inhibition of trehalose synthesis a promising avenue for pest management research. In this study, homology alignment and phylogenetic analysis showed that *FoTPS* exhibited a high sequence homology with TPSs of other insects, including *A. gifuensis*, *T. dendrolimi*, *N. lugens*, *Gampsocleis gratiosa*, and *Locusta migratoria manilensis* (Appendix A). RNAi-mediated *TPS* knockout significantly affects the development and survival of *A. gifuensis* [40], *N. lugens* [29], *D. melanogaster* [26], *M. domestica* [41], *B. minax* [42], and *T. dendrolimi* [43]. Based on these results, *FoTPS* might play a significant role in the survival and development of WFTs.

The direct feeding of synthesized dsFoTPS increased WFT mortality at different developmental stages and significantly down-regulated *FoTPS* expression in first- and second-instar nymphs and adults, aligning with *TPS*-knockout phenotypes in other insects [34,44]. *FoTPS* expression in WFTs was highly elevated during the second-instar nymph and prepupa stages, critical developmental periods from the nymphal stage to adulthood. Moreover, WFT adults require an increased synthesis of trehalose to sustain flight and feeding activities. The enhanced production of trehalose also promotes chitin formation, which is essential for maintaining the rigid structure of the new cuticle [34,45]. Interestingly, the oral delivery of dsFoTPS inhibited WFT egg-laying behaviors, and *FoTPS* was highly expressed during the egg stages, suggesting that *FoTPS* may play a role in the synthesis of egg chitin within the embryonic cuticle of WFTs. Some functional studies on *TPS* in *D. citri* [34], *B. dorsalis* [46], and *Plutella xylostella* [47] have also suggested the involvement of TPS in oviposition. The expression levels of *FoTPS* were lowest in first-instar nymphs and pupae, but highest in adults, a pattern consistent with the temporal and spatial expression of *TPS* in *B. dorsalis* [48]. Therefore, we speculate that the adult stage may respond to the energy demands for mating behavior and reproductive development. However, similar responses were the weakest in first-instar nymphs and pupae.

The oral delivery of dsRNA is the most widely utilized method for inducing RNAi in thrips [49]. However, this approach requires dsRNA to pass through the gut lumen environment, where a significant challenge lies in the presence of gut nucleases [50]. Studies have demonstrated that due to the degradation by these nucleases [7], the ingestion of small amounts of dsRNA is often insufficient to induce rapid mortality in WFTs [14]. To address this limitation, the microbially mediated oral delivery of dsRNA has emerged as a promising strategy, offering a more effective and sustainable approach for RNAi-based control of WFTs [18]. The bacterially mediated RNAi effect mirrors the sublethal effect of low-dose insecticides on WFTs [51], adversely influencing growth, development, survival, and reproduction. In this study, feeding on the dsFoTPS-expressing *E. coli* strain HT115-TPS resulted in significantly higher nymph mortality in the WFTs compared with the control, which is consistent with the effects observed in synthesized dsFoTPS treatments. However, this does not lead to rapid death in WFT adults, although continuous feeding can shorten their longevity. Additionally, a prolonged TPOP, a shortened oviposition period, and a reduced adult oviposition peak collectively led to decreased adult fecundity and lower egg production efficiency. High *TPS* expression in the prepupal stage of WFTs may promote trehalose synthesis to meet the energy demands for chitin synthesis and adult emergence [43]. The key role of chitin in eggshell strength suggests that inhibition of its synthesis could reduce adult fertility as well as egg-hatching abilities [52]. dsRNA-mediated *TPS* silencing down-regulated the expression level of genes related to chitin biosynthesis in *Acyrthosiphon pisum* [30], *N. lugens* [29], and *Tribolium castaneum* [53], potentially regulating chitin synthesis by affecting glycogen phosphorylase and synthetase gene expression. Insufficient glycogen accumulation in adults can affect embryonic development and reduce fecundity [30,53]. Thus, based on population-scale evaluation, the ingestion of FodsTPS can suppress WFT population growth, indicating that *TPS* is an excellent target gene for RNAi-based WFT control.

As an aggregative insect, the WFT frequently forms clusters on infested plant leaves and fruits, and life-table parameters based on population growth provide a more accurate assessment of aggregative insect characteristics under environmental stress conditions [54]. Therefore, developing a robust method to evaluate the overall inhibitory effects of RNAi at the population level is essential [55]. Assessing RNAi impacts on population dynamics can mitigate biases arising from individual variability [56]. Recent studies have assessed dsRNA-silencing effects on population growth in *Plutella xylostella* [57] and *Aphis gossypii* [58], offering deeper insight into the potential of RNAi-mediated pest control strategies. Continuous feeding with *E. coli*-expressed dsTPS resulted in significantly lower *R*_0_, *λ*, and *r* and a shorter *T* than the control. Significantly higher nymph mortality rates and low adult egg-laying capacities were the main causes of lower *R*_0_, leading to decreased reproductive value and life expectancy. The simulated population growth from 10 eggs for 100 d revealed that the population size of WFTs fed with *E. coli* expressing dsFoTPS was only 1/34th of that of the control. This population density was significantly below the economic threshold established for WFT infestations across various crops [59], achieving a level of suppression comparable to that attained through conventional chemical pesticide interventions [5,60]. These results suggested that the continuous oral delivery of *E. coli*-expressed dsFoTPS significantly suppresses WFT population growth, offering an effective RNAi control strategy.

## 5. Conclusions

This study demonstrated that the RNAi of *TPS* significantly increased nymph mortality and inhibited adult reproductive rates, effectively suppressing WFT population growth; this indicates that *FoTPS* is a potential target gene for WFT control and that *E. coli* expressing dsFoTPS facilitated spray-induced gene silencing for the control of WFTs, offering a novel approach and rationale for the prevention and control of WFTs.

## Figures and Tables

**Figure 1 insects-16-00614-f001:**
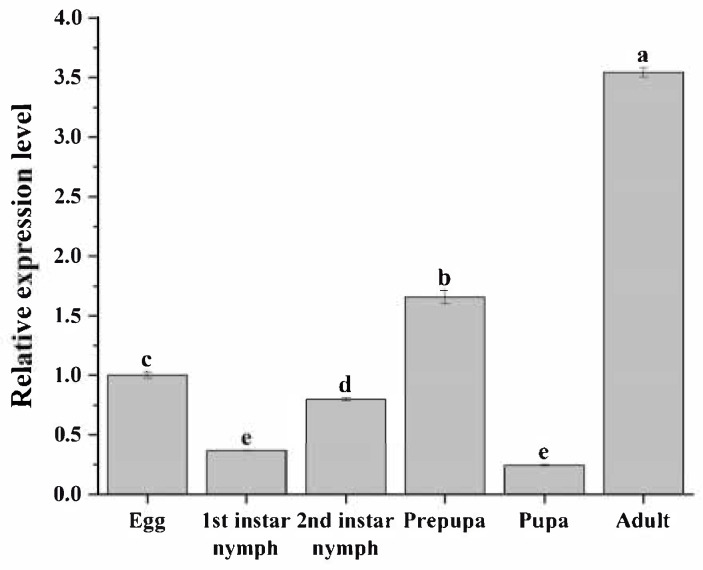
Relative expression levels of *FoTPS* at different developmental stages of *Frankliniella occidentalis*. Bars indicate mean ± SE. The data were analyzed by a one-way ANOVA followed by Tukey’s HSD. Different lowercase letters on the bars indicate significant differences (*p* < 0.05).

**Figure 2 insects-16-00614-f002:**
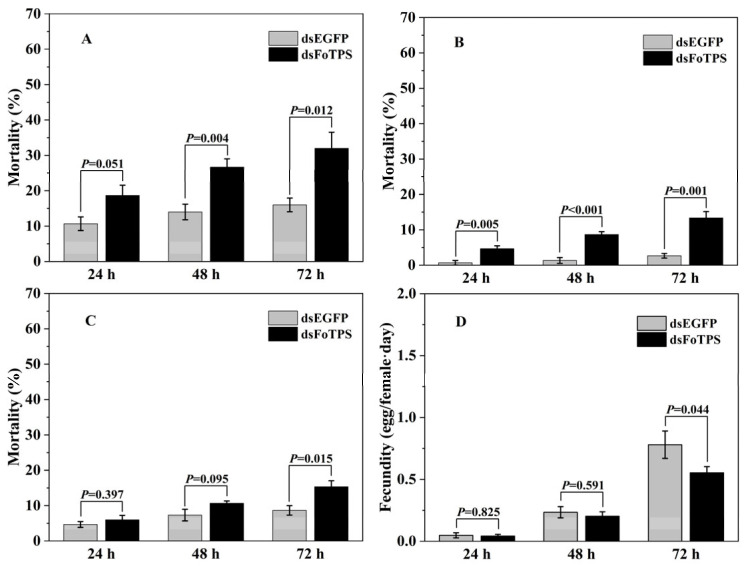
Effects of silencing *FoTPS* on the mortality and fecundity of *Frankliniella occidentalis* fed with in vitro-synthesized double-stranded RNAs (dsRNAs): (**A**) mortality of 1st-instar nymph; (**B**) mortality of 2nd-instar nymph; (**C**) mortality of adults; (**D**) fecundity of adults. Data represent mean ± SE, and *p* < 0.05 indicates statistically significant differences (Student’s *t*-test).

**Figure 3 insects-16-00614-f003:**
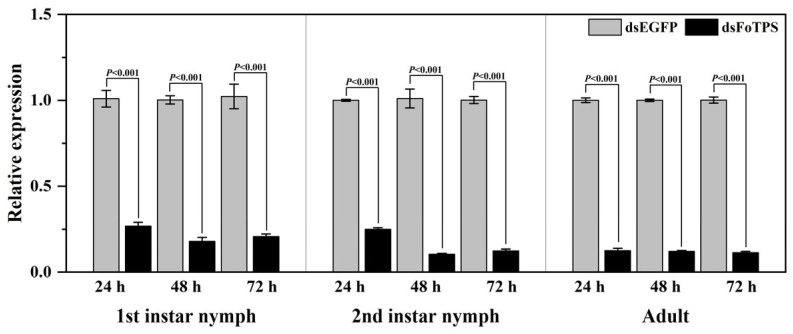
Relative expression of *FoTPS* at 24, 48, and 72 h of feeding with in vitro-synthesized dsFoTPS. Data represent mean ± SE, and *p* < 0.05 indicates statistically significant differences (Student’s *t*-test).

**Figure 4 insects-16-00614-f004:**
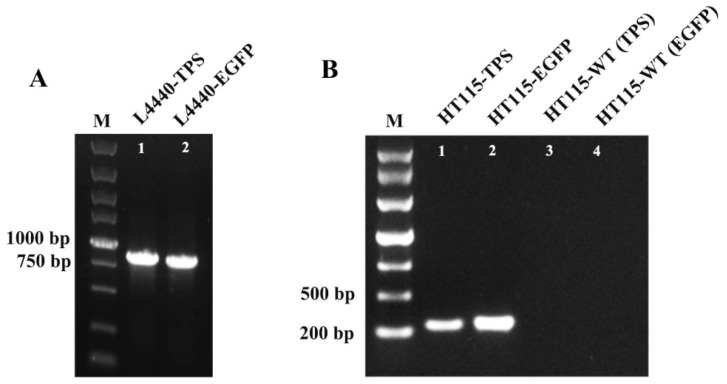
Generation of *Escherichia coli* (HT115) strains expressing dsFoTPS and dsEGFP. Lane M, marker. (**A**) Colony PCR analysis for recombinant L4440; (**B**) induction of HT115 strains to express dsFoTPS and dsEGFP, with HT115-WT denoting the wild-type HT115 strain that was not transformed with L4440, and the parentheses indicating the detection of *FoTPS* and *EGFP* fragments by the wild-type HT115 strain.

**Figure 5 insects-16-00614-f005:**
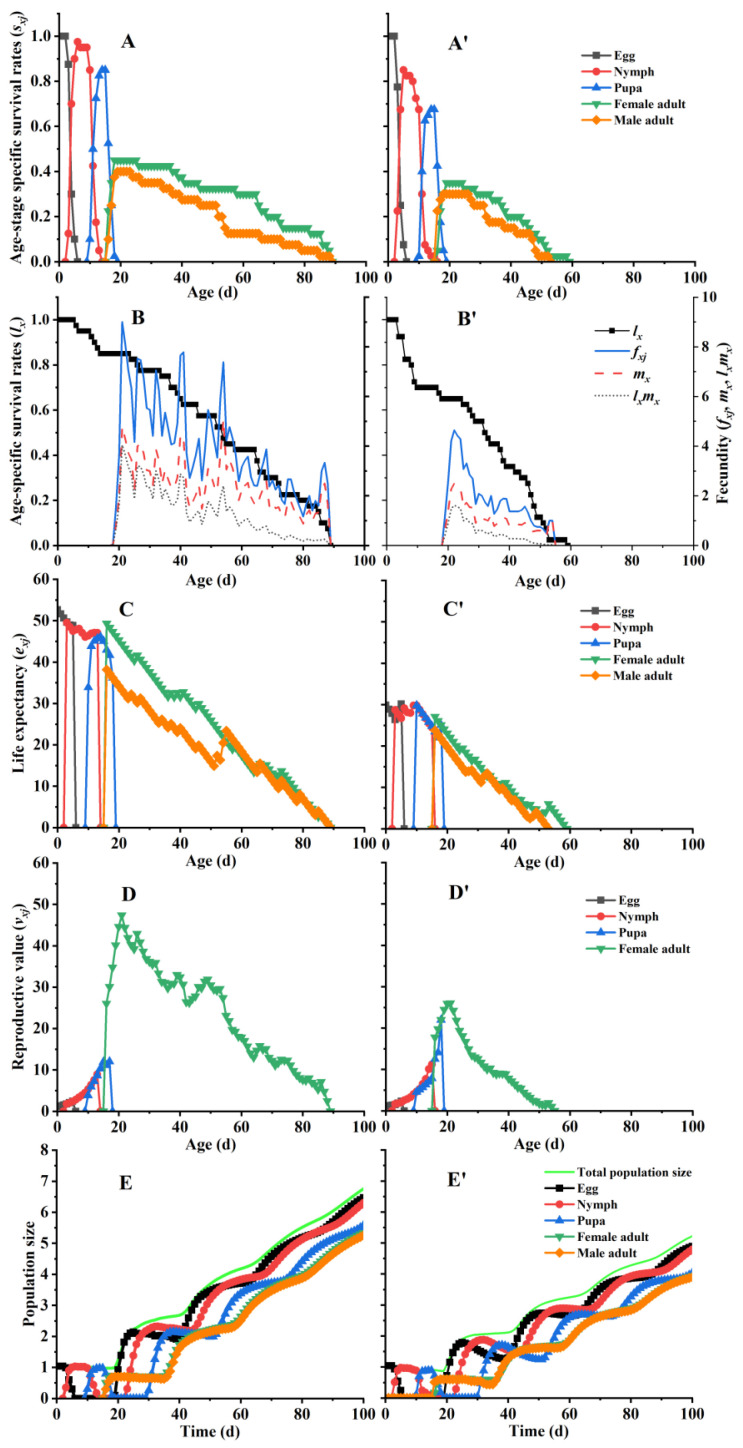
Effects of *Escherichia coli* (HT115) expressing dsFoTPS on the life-table parameters of the *Frankliniella occidentalis* population. (**A**,**A’**) Age-stage-specific survival rates (*s_xj_*); (**B**,**B’**) age-stage-specific survival rates (*l_x_*), age-stage-specific fecundity (*f_xj_*), age-stage-specific fecundity (*m_x_*), and net maternity (*l_x_m_x_*); (**C**,**C’**) age-stage-specific life expectancy (*e_xj_*); (**D**,**D’**) age-stage-specific reproductive value (*v_xj_*); and (**E**,**E’**) total population size. (**A**–**E**) *E. coli*-expressed dsEGFP; (**A’**–**E’**) *E. coli*-expressed dsFoTPS.

**Figure 6 insects-16-00614-f006:**
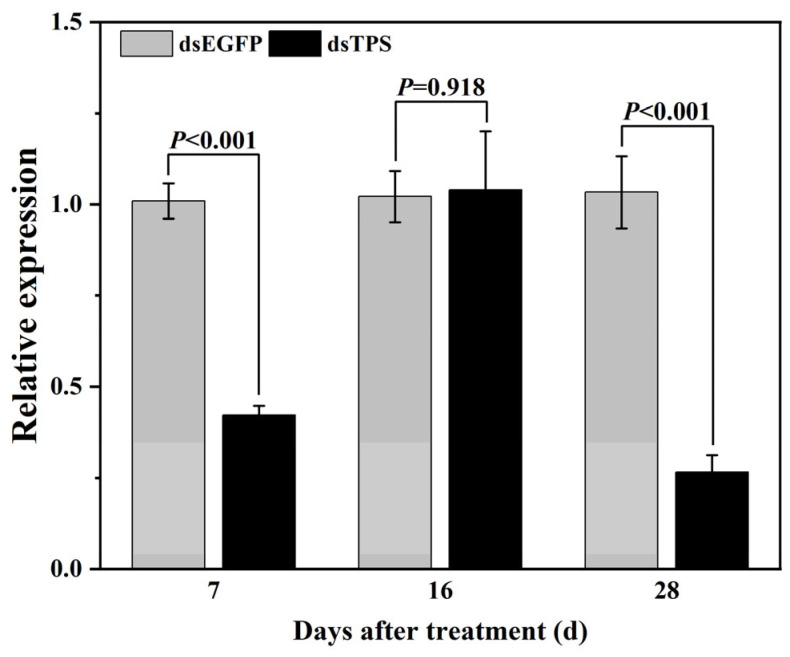
Effects of *Escherichia coli* (HT115) expressing dsFoTPS on the relative expression of *FoTPS* in *Frankliniella occidentalis*. Data represent mean ± SE, and *p* < 0.05 indicates statistically significant differences (Student’s *t*-test).

**Table 1 insects-16-00614-t001:** Effects of *Escherichia coli* (HT115) expressing dsRNA on life-table parameters of *Frankliniella occidentalis* population.

Stage (Parameter)	*n*	HT115-EGFP	*n*	HT115-TPS
Duration of egg/d	40	4.3 ± 0.1	37	4.1 ± 0.1
Duration of nymph/d	38	7.4 ± 0.3	28	7.3 ± 0.2
Duration of pupa/d	34	5.4 ± 0.1	26	5.2 ± 0.2
Duration of preadult/d	34	16.9 ± 0.2	26	16.9 ± 0.2
Adult pre-reproductive period, APOP/d	18	2.1 ± 0.2	14	1.9 ± 0.3
Total pre-reproductive period, TPOP/d	18	18.8 ± 0.2	14	19.3 ± 0.1 *
Total female longevity/d	18	65.3 ± 4.7	14	43.1 ± 2.6 *
Total male longevity/d	16	54.2 ± 4.9	12	39.8 ± 2.6 *
Age of 50% survival/d	40	54.0 ± 6.5	40	31.9 ± 4.9 *
Oviposition days/d	18	45.5 ± 4.4	14	23.3 ± 2.4 *
Fecundity	18	209.2 ± 16.7	14	54.1 ± 4.2 *
Peak reproduction value	40	47.4 ± 2.0	40	26.0 ± 1.5 *
Age of peak reproduction value/d	40	21.0 ± 0.3	40	21.0 ± 0.5
Intrinsic rate of increase, *r*	40	0.1448 ± 0.0071	40	0.1074 ± 0.0091 *
Finite rate of increase, *λ*	40	1.1558 ± 0.0082	40	1.1134 ± 0.0101 *
Net reproductive rate, *R*_0_	40	94.1 ± 18.1	40	19.0 ± 4.3 *
Mean generation time, *T*/d	40	31.4 ± 0.6	40	27.4 ± 0.4 *

The nymph duration included the developmental duration of the first- and second-instar nymphs; the pupa duration included the developmental duration of the prepupa and pupa; and the preadult duration included from egg to pupa before emergence. Data in the table are mean ± SE. An asterisk (*) indicates that the mean in the same row is significantly different at the 5% significance level (paired bootstrap test; 100,000 resamples).

**Table 2 insects-16-00614-t002:** Effects of *Escherichia coli* (HT115) expressing dsRNA on the mortality distribution of the *Frankliniella occidentalis* population.

Stages	*n*	HT115-EGFP	*n*	HT115-TPS
Egg	40	0.000 ± 0.000	40	0.075 ± 0.042
Nymph	40	0.050 ± 0.034	40	0.225 ± 0.066 *
Pupa	40	0.100 ± 0.047	40	0.050 ± 0.034

The nymph duration included the developmental duration of the first- and second-instar nymphs, and the pupa duration included the developmental duration of the prepupa and pupa. Data in the table are mean ± SE. An asterisk (*) indicates that the mean in the same row is significantly different at *p* < 0.05 (paired bootstrap test; 100,000 resamples).

## Data Availability

The original contributions presented in this study are included in the article/Appendix A. Further inquiries can be directed to the corresponding authors.

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
