# Peer review of "Effects of Orally Delivered Double-Stranded RNA of Trehalose-6-Phosphate Synthase on the Population of Frankliniella occidentalis"

_insects, 2025, doi:10.3390/insects16060614_

Round 1

Reviewer 1 Report

Comments and Suggestions for Authors

The authors interfered with TPS, a gene transfering glucose from UDP-glucose to glucose-6-phosphate, in the western flower thrips. They used bacteria to deliver dsFoTPS and tested many life table parameters, including the survival rates. Continuous oral feeding of dsFoTPS inhibits the survival, fecundity and population growth of western flower thrips, which indicates that RNAi targeting FoTPS would be a potential method for thrip management.

This study is interesting, but there are some problems should be addressed.

  1. Give a brief introduction of the function of TPS in the abstract.
  2. Figure 1, it is suggested that the annotation in Figure 1 ("Cloning of TPS PCR product into the destination vector") is written in regular script, not at a tilt.
  3. In Materials and Methods, how many animals are used for qPCR analysis?
  4. In 2.6 Construction of Life Table of WFT Population, “Ten newly-laid eggs of the WFT were then selected” and “Each treatment was repeated four times”, which means there are 40 animals at most in each analysis. In Table 1, the authors got 18 females and 16 males in the HT115-EGFP control group. As in the duration of pupa, the number is 34, so the ratio of male thrips would be 47% and male:female is almost 1:1. I wonder in which condition, people could get this ratio. Write it explicitly in 2.1. Insect Rearing that this rearing condition would get 1:1 males and females.
  5. For female and male longevity test in Table 1, the total number is far from sufficient. n is usually more than one hundred in longevity test. These results are not reliable.
  6. The stages of thrips include prepupa and pupa, but in Table 1, “duration of pupa” and “duration of preadult” are ambiguous. The authors have to use the same statement.
  7. Figure 3, what is the age of females for fecundity analysis. The time in x-axis is missing in 3D. The P value of significant difference has to be shown in the figure legends of Figure 3 and Figure 7.
  8. Line 346–351, one could not read the data in Figure 6. The authors could give another histogram to show it.
  9. dsFoTPS affects many life table parameters, which could induce mortality of thrips. The authors would discuss in Discussion how TPS affects these parameters, leading to thrip death.

Reviewer 2 Report

Comments and Suggestions for Authors

The manuscript "Effects of orally-delivered double-stranded RNA of trehalose-6-phosphate synthase on the population of Frankliniella occidentalis" is an interesting study showing that the population of this insect can be suppressed by silencing the trehalose synthase gene using E. coli that expresses dsRNA targeting this gene.

The manuscript is well written, and the experiments are well-designed. However, I suggest minor changes to improve its readability.

-Figure 1 can be moved to supplementary material, since it is not a new method and is not part of the results.

-The Supplementary Figures 1 and 2 legends have to include more information about subfigures, since all this information is missing. 

-In Figure 7, the authors hypothesized that no silencing at 16 days is most likely because the pupa instar cannot feed. In addition, they commented that constant exposure is required to reach gene silencing in several species. This result is well discussed, and I wonder if the authors have measured the prevalence of E. coli throughout the life cycle of the insect. This is not a required experiment, but it could provide insight into the duration and consistency of dsRNA exposure using the E. coli delivery system. This is particularly relevant given that E. coli is not a natural insect symbiont; its use for RNAi delivery generally requires continuous supplementation or genetic engineering to enable stable gut colonization. Exploring this aspect would strengthen the model's applicability.
